# Effects of Advertising: A Qualitative Analysis of Young Adults’ Engagement with Social Media About Food

**DOI:** 10.3390/nu13061934

**Published:** 2021-06-04

**Authors:** Annika Molenaar, Wei Yee Saw, Linda Brennan, Mike Reid, Megan S. C. Lim, Tracy A. McCaffrey

**Affiliations:** 1Department of Nutrition, Dietetics and Food, Monash University, Notting Hill 3168, Australia; annika.molenaar@monash.edu (A.M.); wei-yee.saw@monash.edu (W.Y.S.); 2School of Media and Communication, RMIT University, Melbourne 3000, Australia; linda.brennan@rmit.edu.au; 3School of Economics, Finance and Marketing, RMIT University, Melbourne 3000, Australia; mike.reid@rmit.edu.au; 4Behaviours and Health Risks, Burnet Institute, Melbourne 3004, Australia; megan.lim@burnet.edu.au; 5Melbourne School of Population and Global Health, University of Melbourne, Carlton 3053, Australia

**Keywords:** food, advertising, marketing, young adults, social media, social marketing

## Abstract

Young adults are constantly exposed to energy-dense, nutrient-poor food and beverages, particularly through advertising. Exposure can influence poor food choices and negatively impact health. This study aimed to understand young adults’ attitudes and experiences associated with food-related advertisements, particularly on social media. This qualitative analysis involved *n* = 166 Australian 18 to 24-year-olds who were involved in a four-week online conversation on different areas relating to health, social media, and eating. Inductive thematic analysis was utilised on two forums on the recall and perceptions of food-related advertisements. Young adults commonly mentioned aspects of the marketing mix (promotion, product, price, and place) in food advertisements. Participants were more readily able to recall energy-dense, nutrient-poor food advertisements compared to healthy food-related advertisements. Digital advertisements were often discussed alongside the use of ad-blockers and algorithms which tailored their social media viewing to what they like. Participants felt constant exposure to unhealthy food advertisements hindered their ability to realise healthy eating behaviours and created feelings of guilt. This current analysis highlights the need to provide an advertising environment that appropriately motivates healthy eating and a food environment that allows healthy food to be the affordable and convenient option.

## 1. Introduction

Food advertising has been found to be an important influence on food choices, eating behaviours, and consumption patterns [1]. Energy-dense, nutrient-poor (EDNP) foods are associated with negative health consequences [2,3]. Additionally, EDNP foods are some of the most advertised foods on television [4,5,6,7] and online [8,9,10]. Increased exposure to fast-food advertisements has been associated with increased consumption of EDNP foods in adults [11]. The effects of food advertising on children and adolescents have also been widely studied due to the relationship between these advertisements and unhealthy dietary choices [12,13]. Consequently, marketing of EDNP foods to children is regulated in many countries [14,15,16]. Research on young adults specifically is more limited, with some evidence indicating potential vulnerabilities to food advertising [17,18].

Young adults have amongst the poorest dietary behaviours compared to other groups, with diets often high in EDNP foods and low in fruit and vegetables [19,20,21]. Young adults aged 18 to 24 years experience a potentially vulnerable and malleable transitional period known as the “emerging adulthood” years [19,22]. During this developmental stage, young adults encounter important life transitions which enable them to gain independence and responsibility, as well as possibly establish long-term health-related behaviours [19]. Although young adults may be less likely than older age groups to have current major health problems, they are still at risk of negative effects of unhealthy lifestyle behaviours which may track into later adulthood, potentially leading to serious health consequences later in life [19]. Young adults are therefore an important age group to understand in order to encourage the establishment and maintenance of healthy behaviours and protect individuals from the effects of EDNP food advertising.

Traditional food advertising channels are television (all types), radio, print (magazines and newspapers), and billboards [23]. However, in the digital era, advertising takes place on social media platforms such as Facebook and Twitter [24,25]. Social media advertising can be defined as “the utilization of social media technologies, channels, and software to create, communicate, deliver and exchange offerings that have value” [26]. Social marketing, which aims to utilise marketing principles in order to create behaviour change for social good [27], also utilises social media [28]. Social media is utilised by both food and health industries as an advertising platform to increase brand and information reach [9,29]. Young adults are amongst the largest users of social media, with Australian 18 to 29-year-olds spending on average the longest amount of time on different social media platforms [30]. Due to the pervasiveness of social media, young adults are also constantly exposed to advertising of EDNP foods and beverages on social media [8,9,18]. Furthermore, social media is frequently used by young adults to communicate, gather, and share food- and health-related information [31,32].

As a result of social media’s effectiveness as a platform to disseminate food and health information [33,34,35], social media advertising has the ability to impact young adults’ health, regardless of whether the information provided is accurate or inaccurate [8,18,34,35]. Social media advertising relies heavily on social influence strategies, harnessing the economic value of social networks and social aspirations to persuade people to buy [36]. Using artificial intelligence, advertisements can be created, executed, evaluated, and refined within hours, thereby enabling more profound levels of persuasion than ever before [37].

An important component of social media advertising is that it can be created and targeted on the basis of algorithms, which track and react to individuals’ participation in the digital environment [24]. Further, this form of advertising is designed to ensure that ads are relevant to the context of participation and are a “natural” extension of peoples’ behaviours [38]. Consequently, advertising may not be automatically discernible as advertising because it is integrated seamlessly into an individual’s lived experience [18]. Such covert advertising is also known as natural or native advertising, but other masked forms of advertising can include sponsorships, influencer marketing, content marketing, stealth marketing, and product placement [39]. This is of concern, as individuals are often unaware that they are being persuaded to purchase EDNP food products [17] and are not able to discern the influence these advertisements are having on their food choices [40]. This is particularly apparent with children, and therefore governments have attempted to mitigate the effect of unhealthy advertisements by introducing regulations [41,42].

All these forms of advertising are designed to create a positive attitude towards a brand and persuade people to buy the brand’s products without necessarily being aware that they are being persuaded. When individuals are involved in or interested in the product, direct (overt) forms of advertising are applicable because they fulfil an information need that leads to people being able to make considered choices about their purchase behaviour. However, covert forms of advertising are applicable when the target audience is not interested in the product or motivated enough about it to pay attention to more complex messages in advertising [43] or they are unwilling to make considered rational choices [44]. In these cases, the indirect route to persuasion involves advertising in such a way as to increase the likelihood of behaviour that might lead to a change in attitudes or beliefs without the cognitive processing of an advertising message [45]. When it comes to food advertising on social media, there is a plethora of messaging using both direct and indirect forms of persuasion [46].

Additionally, when it comes to eating healthfully, young adults have been found to be overwhelmed with contradictory nutrition information on social media [47]. The exposure to this potentially untrue and contradictory food and health information may cause cognitive conflicts and interrupt active decision making for healthy eating [47]. This is particularly problematic in social media advertising for food products where a majority of information or advertising is presented with simple claims and enticing visuals [48]. Unhealthy food advertisements have been shown to have a significantly more positive response by adolescents compared to healthy food and non-food-related advertisements [49]. The ability for EDNP food manufacturers to persuasively advertise their food products to young adults and create a digital environment which features EDNP foods presents a serious concern for public health and the diets of young adults. Therefore, it is important to understand what food related advertisements young adults are seeing and what effect these advertisements have on their attitudes and behaviours. Therefore, this study aims to understand young adults’ attitudes and experiences associated with food-related advertisements, particularly online and on social media.

## 2. Materials and Methods

### 2.1. Online Conversations

This study reports findings from the first Phase of the “Communicating Health Study” [50]. The Communicating Health study, which runs over four years, aims to apply social marketing techniques to understand how young adults engage with social media regarding both health and healthy eating in order to guide the development of effective social marketing messages that motivate and engage young adults. Communicating Health Phase 1a was a formative research stage which involved a four-week online conversation with young adults discussing a series of health-, social media-, and eating-related topics. Online conversations are a market research methodology that incorporate the principles of digital ethnography [51,52] and involve the use of an online community methodology which gathers rich insights into consumer behaviour [53]. Online conversations involve communities of participants who are able to discuss different topics over a longer period of time in comparison with other qualitative methods such as focus groups and interviews.

This study received ethics approval from the Royal Melbourne Institute of Technology Business College Human Ethics Advisory Network (project number: 20489) and Monash University Human Research Ethics Committee (project number: 7807).

### 2.2. Recruitment

The study aimed to recruit 200 young adults aged 18 to 24 years old. This target was based on previous work using a similar methodology [54] in order to achieve an extensive amount of data exploring the topic area. Participants were recruited by an Australian Market and Social Research Society-certified field house [55]. Young adults who had previously consented to participate in research with three research panels from across Australia (Survey Sampling International (SSI) https://www.surveysampling.com/ accessed on 27 March 2019, Pure Profile https://www.pureprofile.com/au/ accessed on 27 March 2019, and Student Edge https://studentedge.org/youth-research-insight-services accessed on 27 March 2019) were invited to participate in this study. Three panel partners were used to ensure a mix of participants from across Australia as well as to reach the target quotas which were to recruit a sample of participants that is approximately representative of the Australian population [56] on location (both Australian State or Territory and location type, i.e., metropolitan and regional locations) and gender. All panels were International Organization for Standardization (ISO)-accredited for the purpose of market and social research [57]. Participants in these panels voluntarily added their names to the field houses’ databases with the expectation they would then be invited to participate in different market research projects.

A screening survey which included demographic information, self-reported weight, height, interest in health, and social media use was sent to panel members to assess their eligibility [58]. Panel members were eligible if they were aged between 18 and 24 years old, self-reported using social media at least twice a day, and were currently residing in Australia. Completers of this survey who were eligible were sent a link to a profiling survey and were asked to register on the online communities’ website. Participants who registered (*n* = 234) were assigned to one of four online communities on the basis of their age and interest in health (four approximately equal online communities based on age: 18–21 or 22–24, and self-rated interest in health: low or mid/high). Interest in health classification was based upon the median value for the following question from the profiling survey: “On a scale of 1–7 where 1 means ‘Strongly Disagree’ and 7 means ‘Strongly Agree’, please indicate how strongly you agree with the following statement—’I take an active interest in my health.’” Participants were separated into four online communities in order to create more homogenous groupings to encourage counterpart conversations; however, data from the four communities were analysed collectively.

Not all participants who completed the profiling survey responded to all of the forums throughout the online conversations. The dropout rate overall (Figure 1) was large, possibly due to the age group and length of the study, which led to the implementation of a referral system, whereby participants could invite their friends (who were then screened and profiled in the same way). While having participants invite friends more closely mirrors social media it may produce some confounding issues, where friends with like views are part of the online conversations. However, the friend referrals were not necessarily placed in the same online community in terms of how they were profiled by their age and interest in health. For participating in the study, participants received an AUD 100 gift voucher as an incentive, and the 20 most comprehensive contributors (i.e., five participants per online community) received an additional AUD 100 voucher.

### 2.3. Data Collection

The online conversations were operated and moderated by an independent market research field house with topics and questions developed by the Communicating Health researchers. Topics and questions were presented through an online “lounge” hosted on a private website over four weeks (10 May to 6 of June 2017). The online conversations included 20 different topic forums and two challenges which took approximately five minutes each to complete (approximately 110 min total), three short polls, and a journal entry that required at least four contributions throughout the four weeks [58]. There were two moderators including both a male (MMgt Marketing/Finance) and female (BA Psych Sociol, MA Applied Social Research) moderator who both had extensive experience in consumer and social research. As moderators were unknown to participants, an introductory forum was used to build relationships between participants and moderators. Forums were opened and different questions were posed at different stages throughout the four-week period. All forums remained open for the four-week period and participants were able to post responses in any of the forums up until the final day. Data collected were in the form of digital text responses and uploaded images in response to the questions posed. For this paper, two of the forums from the online conversations were analysed (discussion guide for each forum found in Table 1). These specific forums were chosen on the basis of their discussion of food advertisements and were therefore the only ones that provided insight into the research question about attitudes and experiences relating to food advertising. Discussion guides for the remaining forums and challenges in the online conversations are available here [58]. Due to the forums running over different weeks, there were different completion rates, with *n* = 166 young adults completing Forum 3 “Ads about food” and *n* = 145 young adults completing Forum 12 “The health ads we notice”.

### 2.4. Data Analysis

Inductive thematic analysis [59] was used to analyse the verbatim online text responses from both forums. The analysis was conducted on the group of participants as a whole, as making comparisons between different groups based on their characteristics was not the aim of this study. To familiarise themselves with the forum text responses, the authors (W.Y.S. and A.M.) independently read through all 369 responses from Forum 3 and all 190 forum responses from Forum 12. Text responses included individual answers to the posed questions as well as interactions between moderators and other participants. The responses were manually inductively coded and grouped into categories of like responses. This process involved reading the data multiple times to gain an understanding of the overall dataset and then coding the data line by line into codes relating to the subject of that line. These individual codes were then grouped into categories of like responses in order to develop themes. The analysis from both forums were interpreted collectively in the development of overall themes despite the topics and forums themselves being discreet. Analysis was focussed on food products, both “healthy” and “unhealthy” mentioned across both forums in line with the aim of the analysis. Investigator triangulation was employed to enhance the rigor of the thematic analysis and as a form of validation [60] where two authors coded the forums separately and grouped their like codes before meeting to compare and contrast their coding and to come to a consensus before developing and refining themes. The analysis of all forum responses was conducted by two female authors with backgrounds in nutrition science (W.S. and A.M.). For anonymity, brand and identifying names are removed from participant quotes otherwise quotes are verbatim.

## 3. Results

### 3.1. Participant Characteristics

The demographics for *n* = 166 participants who completed at least one of the forums analysed (Forum 3 or Forum 12) can be found in Table 2. Overall, 60.8% of participants were female, 53.0% were of a healthy weight, and 74.7% spoke English as their primary language at home. Participants were generally residing in an Australian Metro location (80.1%), with 48.2% currently living with their parents. Many participants were currently studying (66.9%) and were primarily completing an undergraduate degree (48.2%). The average overall disposable weekly income was low with 39.2% of participants having less than AUD 40 a week.

### 3.2. Thematic Analysis

During triangulation of findings, the authors had similar themes related to the promotion of products and price. The authors believed these themes fit with the marketing mix or the 4Ps of marketing framework that is an established framework in marketing [61]. Therefore, the 4Ps framework was used to guide the discussion of the results from the participants. The four major themes represented therefore were: Theme 1—promotion, Theme 2—product, Theme 3—price, and Theme 4—place (Table 3).

#### 3.2.1. Theme 1: Promotion

Nearly all participants believed they were well aware of the products that were being promoted to them in their everyday lives. As they were aware that products promoted to them were targeted on the basis of their search history and interests, the advertisements they described were generally of products they were interested in. When asked about health-related advertisements, some participants knew they could not recall such advertisements as they are not interested in health products, so those companies would not be targeting promotions towards them.


*“The ads on my social media are tailored to suit what I would like to see, and the internet knows I don’t want to see health ads. I mostly see ads for website creation tools, online stores that I frequent and video games.”*
(Forum 12: Male, 21 years old)

While many were seemingly aware of the advertising environment they were in, some did not realise the effects the advertisements were having on them, particularly those advertisements that were more covert. Some believed they could “tune out” any product promotions and therefore believed they never really saw advertisements. However, these participants who stated they did not really see advertisements were still able to remember and recall different advertisements when prompted in the forums.


*“Hey Y’all, I also saw the new [Brand name removed] chocolate block—it really grabbed my attention. I think using bright colours will always get people’s attention and I don’t even fancy chocolate that much. I think of myself as ad- impenetrable. I often won’t buy products specifically because they are being advertised”*
(Forum 3: Male, 22 years old)

Some participants stated that they use ad-blocking services so they do not have to see the large amounts of advertising on online platforms. However, these individuals were still able to recall example promotions, indicating even with ad-blocking services they were still faced with advertisements at some point or on some platforms.


*“It could be I’ve just grown to tune them out. Oh, something interesting to note also, I do have an adblocker installed, but I turn it off for sites I like (such as facebook and youtube), it’s there purely to protect me from annoying sites or ones which load slowly due to the amount of ads on it.”*
(Forum 3: Male, 23 years old)

##### Promotion Strategies That Worked

There were certain aspects of the food promotion that appeared to commonly grab and maintain the attention of these young adults. Frequency and repetition of an advertisement was effective in gaining the attention and sticking in the memory of these young adults.


*“As many have already said, the [fast food brand name removed] ad comes to my mind first. The new [fast food product name removed] ad... has come on the many times that my mouth waters and I get the urge to go past [fast food brand name removed] hahaha.”*
(Forum 3: Female, 20 years old)

Participants were more likely to enjoy when promotions were of food products and services that were relevant to their needs and interests. Moreover, promotions for new products that were related to, or similar to, products they already liked were of interest and something they could see themselves consuming. A combination of high frequency and foods that were relevant and appealing to them was described as a catalyst for purchasing a particular food.

Promotions that were visually appealing, made the food look delicious, and used “happy” or bright colours were mentioned commonly.


*“I think this was down to the colours. The ad used pastel colours which made it very easy on the eyes so it was eye catching and pleasant to look at—I wanted to stop and see what the ad was for”*
(Forum 3: Female, 21 years old)

Other promotion strategies that were attention-grabbing included interactive promotions such as competitions or opportunities to win a reward or voucher to save money on the food. Jokes, wit, slogans, and jingles were attention-grabbing and memorable parts of food promotions. Also mentioned was promotion by celebrities and influencers. Some participants indicated that, at times, this encouraged them to try the products or make changes to their current lifestyle to live up to the outcome advertised by this celebrity or influencer.

##### Promotion Strategies That Were Disliked

However, advertisements were not always received favourably. Participants stated that advertisements that promoted deals were sometimes viewed as deceptive rather than the good value for money that was intended. Some participants were sceptical that special deals actually saved money and believed it was more likely to lure them in but then they would spend more than intended.


*“Creating ‘meal deals’ is one of the best things fast food places have done. It makes people feel as though they are getting a good deal, while walking away having spent more money and eating more calories than they initially planned.”*
(Forum 3: Female, 24 years old)

Participants wanted advertisements that were tailored to their interests, so when they did happen to see products being promoted to them that they had no interest in, this was seen as annoying. Some jingles and repetitive aspects of product promotion were also viewed as annoying, despite them being “catchy” and even, in some cases, influencing behaviour.


*“I guess that even if it’s annoying, the fact I remember it suggests that it worked. I even shop at [brand name removed], and recognise some of the specials.”*
(Forum 3: Male, 22 years old)

Mostly, seeing an advertisement very frequently was not viewed positively, despite it being successful in being remembered by participants. High frequency promotion was viewed as “annoying”, sometimes even considered “aggressive”, and generally decreased the likelihood that they would go and purchase the advertised food.


*“It played sometimes 3 times an ad break. Once I saw it back-to-back twice. It was exhausting.”*
(Forum 3: Female, 24 years old)

#### 3.2.2. Theme 2: Product

These young adults frequently described EDNP food products, with fewer participants recalling advertisements for healthy foods. Even when asked about health product advertising, participants often described unhealthy EDNP food or fast food instead. There was a sense of familiarity and affinity towards a lot of the foods being described, with participants stating they already ate and enjoy these foods. Knowing how the food tastes sometimes meant the participants were more enticed by the advertisement.


*“I think it’s easier for food chains (especially the popular ones) to grab our attention because we already know what we’re getting, most of us would have tried their food before and know how delicious it is. so going back for a special deal or to try something new is motivating for us.”*
(Forum 3: Female, 22 years old)

Participants often discussed the perceived healthfulness of the foods being advertised to them. Some participants communicated their discontent with being constantly exposed to fast-food advertisements, as these advertisements also hindered their ability to make healthy choices.


*“I’m trying to eat healthier and all I see is fast food around me. That really makes it difficult to stay motivated and to avoid derailing back into unhealthy takeaway option.”*
(Forum 3: Female, 18 years old)

The inability to consume a healthy diet, as the participants thought they should, often gave rise to feelings of guilt and shame as they mentioned they crave/could not resist “unhealthy” foods.


*“Most of the food ads I get are for fast food or delivery services, and I try to hide them when I see them so that I can avoid temptation!”*
(Forum 12: Female, 24 years old)

Some participants indicated that having more healthier food choices advertised that satisfied their needs for convenience and price and less advertisements for EDNP food might ease the process of making healthy choices.


*“I have to agree with the [EDNP food brand name removed] ad, that is the only ad I have really taken notice of as well. It’s a little bit disheartening when you think about it that unhealthy foods are the ones that get the spotlight. I’d love to see the 5&2 ads come back!”*
(Forum 3: Female, 22 years old)

#### 3.2.3. Theme 3: Price

Participants often reported noticing advertisements that highlighted affordable deals, discounts, and special offers. These participants often described themselves as students or on a low income and therefore the affordability of food and budgeting were major factors in their purchasing behaviours.


*“On a budget as a university student the only ads that catch my attention are either cheap or on a good deal, like ads about [fast food brand name removed], [fast food brand name removed], [fast food brand name removed] etc. I am not particularly happy with my food choices, but eating healthy can’t be achieved by comparatively expensive healthy options to unhealthy ones.”*
(Forum 3: Male, 19 years old)

Value for money often caught the eye of the participants as it was relevant to their needs. Price tempted participants to want whatever the advertisement was selling, and some participants stated price was the ultimate reason for them purchasing the advertised food. Even when specifically asked to recall advertisements about health, discounts and value for money of fast food and EDNP food were mentioned.


*“With all the promos and deals that most junk food restaurants offer, they make it seems like a big bargain which sometimes it is, so it makes you want to go out and buy it.”*
(Forum 3: Male, 21 years old)

There were opposing opinions throughout the forums regarding the cost of convenience and takeaway foods in comparison to buying and preparing fresh foods from a supermarket. Some participants believed that buying and cooking their own food would work out cheaper, while others believed convenience and takeaway foods were cheaper, more readily available, and suitable for their busy lifestyles.


*“I dislike these [fast food] ads because they are misleading and don’t offer anything positive. Often working class people feel as though these foods are all they can afford, due to dollar menus and $5 meals, however it is consistently shown that whole foods are cheaper in the end.”*
(Forum 3: Female, 24 years old)

While value advertising strategies were appealing, and initially sounded like a cost-saving purchase, participants described how they usually end up buying more. This was especially mentioned in relation to fast-food companies with “meal deal”-type promotions.


*“The marketing they are using is to get people to visit the restaurants, people like myself are unlikely to spend only $3 because we concider this a good deal, it is likely that the more gullible of us (like myself) will spend more money than intended at the restaurant.”*
(Forum 3: Female, 24 years old)

The food products and services mentioned were most commonly of the convenient or ready-to-eat variety. Even the healthy food options mentioned were mostly convenient health foods or healthy food delivery services rather than whole foods. A perceived lack of time to prepare food drove participants’ desire for convenience foods. Some participants also stated they had no desire or know-how to cook food so instead opted for ready-made food or food delivery.


*“Everyday I see this one [food delivery service brand name removed], first time I seen it got me interested for the convenience of the service. Ticked all my boxes healthy, easy, fast and not to much worry about planning.”*
(Forum 12: Female, 24 years old)

#### 3.2.4. Theme 4: Place

When asked to recall an advertisement, most participants stated they see them passively when scrolling through social media. Participants rarely stated they were following the food brand of whose advertisement they were discussing. Facebook and YouTube were the most common social media platforms mentioned for seeing advertisements. Healthy food-related advertisements identified on social media were generally presented on Facebook, Instagram, and YouTube.


*“I’ve seen the 2 fruit 5 veg ad on Facebook multiple times. Definitely something that I find intriguing and occasionally try to do, but not always successfully.”*
(Forum 12: Male, 23 years old)

Despite being asked about online and social media advertising, participants commonly mentioned advertisements seen on traditional media, particularly on television, radio, and billboards.


*“I have recently seen many ads on [fast food brand name removed] mainly on facebook, i find it very annoying i also see it a lot on tv I never really see healthy food ads on tv or in any media i use daily.”*
(Forum 3: Female, 22 years old)

Some participants described the overt traditional media advertising as more likely to catch their attention as they were not able to “skip” past it. However, some participants described social media advertising as more persuasive and preferred this more covert advertising, for example how Instagram incorporates advertisement into the feed more naturally.


*“One of things I like about advertising on Instagram is that is very non-intrusive. It blends in with the feed very nicely. I believe that ads that are placed in Instagram are generally more reputable mainly because they are often accompanied with abosultely amazing shots of the food. It gives you a reason to like it. Also the fact that it is disguised as a post, instead of an ad is quite nice.”*
(Forum 3: Male, 23 years old)

Also important to participants was the place in which the product advertised could be purchased. Products advertised that could be purchased in convenient locations, such as ordering online and having food delivered to you or picking food up on the way home or near university, were appealing to many participants. Convenience was key to grabbing the attention of many of these time-poor young adults.


*“Yes [brand name removed] ads seems to really persuade you. I always get persuaded with the [brand name removed] though. It’s nice to know I can get my donuts near my home instead of driving miles away for it.”*
(Forum 3: Female, 22 years old)

## 4. Discussion

This qualitative analysis explored Australian young adults’ experiences related to food advertisements. Elements relating to the promotion, product, price, and place of food advertisements were often discussed as aspects that increased the appeal and recognition of the advertisement. The use of algorithms to target native advertisements that bypass ad blocking software was also prevalent, although our young adults were often able to assess when they are being advertised to. Advertisements that would increase product related intentions were those that promoted the cost (affordability), convenience, and relatability of the product or even the advertisement itself. Companies that used frequent advertising were easy to recall but at the cost of being considered aggressive and reducing the appeal to buy those foods. Participants also suggested that the ability to eat healthfully is very difficult and led to feelings of guilt, as the marketing environment limits the ability to avoid unhealthy alternatives.

### 4.1. The Marketing Mix

Advertising, whether digital or traditional, supports a combination of elements of a marketing mix [62]. The marketing mix is sometimes known as the 4Ps of marketing (product, price, place, and promotion). While there are arguably other elements in marketing [61], the 4Ps were used a framework for the results of this paper as they were evident within the participants’ responses. In the context of social media advertising, the product can be an idea or information as well as any tangible good. It is the object that is at the core of the exchange. Price is the cost associated with adopting the product. Costs can be non-monetary such as time, effort, and consequences, and includes issues such as affordability and accessibility. This was evident in the desire for convenience expressed by participants. Some of these young adults believed they did not have the time or skills to consume healthy food and therefore the cost of purchasing convenient EDNP foods was more appealing. Place is where individuals take up or access the product as well as how they will be reached or engaged. This definition goes beyond geographic location and logistics and includes the types of social media and traditional media platforms mentioned by participants. Promotion is anything associated with getting a message from a sender to a receiver and includes both covert and overt communication techniques.

An alternative application of the marketing mix than promotion of EDNP food is its application in social marketing to positively influence health-related behaviours [63,64,65]. Successful adaptation of the marketing mix in healthy eating social marketing interventions has been shown to affect positive behaviour change, particularly when compared to using just communication strategies alone [66]. In the current study, advertisements that implemented multiple components of the marketing mix were noticed by participants to a greater extent, particularly those used in EDNP food advertisements. This could be because health-related advertising tends to use single P marketing and rely heavily on information/appeal promotion [67]. EDNP food advertising, on the other hand, evokes an array of emotional, sensory, and appetitive appeals of the product to promote intentions to engage [68].

### 4.2. Advertising on Social Media

Participants were less likely to discuss seeing advertisements on social media, despite large amounts of advertising expenditure in Australia being devoted to online advertising [69]. This could be due to the covert nature of social media advertising or social media advertisements being ignored or blocked by participants. It may also be that attention is not paid to advertising on social media, which is different to seeing and then ignoring ads, especially in the fast-paced world of social media where scrolling and swiping are prevalent behaviours [70]. Participants reported passive consumption of social media advertising (i.e., seeing but not engaging with the advertisement); however, the constant exposure to food advertisements received primarily negative feedback. While the number of advertisements is intrusive, impressions, exposure, and reach remain key metrics for social media effectiveness [71]. Consequently, there is unlikely to be a decrease in advertising over the coming years. Furthermore, social media is used by food advertisers to normalise products and increase product attractiveness to consumers regardless of the health implications [17,72].

Automatically sourced content and user-generated content are widely used on social media platforms to influence consumer engagement by embedding and tailoring advertisements on the basis of consumer interests [8,73,74]. Participants stated that advertising through social media had a greater persuasive impact and initiated more positive intentions than television advertising, even though more attention was paid to television advertising. This is consistent with prior studies that have found that consumers are less aware of the persuasive intent behind social media advertising [75]. The results of this study also support the contention of Dunlop et al. [8] who found that consumers underestimate their degree of exposure to advertisements on social media and are less likely to view social media advertisements as a form of advertising. Marketers have taken to using the power of peer-to-peer transmission of content on social media as a form of persuasion to purchase their unhealthy products, which is generally not distinguishable by young people as advertising [76]. Most participants described more covert forms of advertising online, which may be due to the need for food marketing to be covert to pass regulatory guidelines [77].

### 4.3. Food Prices and Affordability

The results also show that the price of products is an important determining factor for food choices, with affordability limiting their ability to access healthy alternatives. Food affordability is a common concern for young adults [22,78], especially when they first gain financial and social independence [19]. Young adults are in a unique life stage where they are learning to budget on their own and are often studying with limited income to support consumption of healthy foods [78]. Previous research has identified affordability as a major driving force for dietary behaviours [79,80], with people often perceiving healthy food as more expensive than unhealthy food, and therefore not something that may be a reality for their lives [80]. However young adults have also indicated that takeaway food is both expensive and unhealthy and that they could improve self-care by reducing the consumption of these foods and consuming more healthy alternatives [81]. To achieve the reduction in takeaway food, they would require a food environment that is supportive to change and reduces their exposure to unhealthy food [81]. Social marketing campaigns or advertising that aim to encourage consumption of healthy food therefore must consider the cost of the healthy alternatives they are promoting and may benefit from drawing attention to the cost per nutrition weight of these foods compared to EDNP foods [80]. Young adults, particularly those in university, are known to prioritise affordability of food and have a focus on budgeting to afford all their living expenses [82,83,84]. Our results also show that price was a key driver of unhealthy food choices, with most participants suggesting that healthy food was unaffordable and therefore often relying on EDNP food, particularly fast food. The food environment and accessibility of healthy foods is a likely contributor to this reliance on EDNP. Living in a food desert or food swamp with limited access to nutritious fresh food and/or greater amounts of EDNP convenience and fast foods have been linked to poor dietary habits and poor health outcomes [85,86].

### 4.4. Covert and Overt Advertising

The inclusion of attractive visuals and audio have been shown to increase the noticeability and attention paid to advertisements by young adults [9,46,63,87]. Young adults are often aware of overt advertising and are able to discern marketing activities that are being directed towards them [17]. On the other hand, covert advertising removes the freedom of choice surrounding whether or not people want to engage with information about a brand or product, as people are not always aware that consumed content is advertising. For example, the majority of U.S. top-grossing movies over 25 years portrayed unhealthy food (73%) and beverages (90%), with branded products accounting for only 12% [88]. However, despite being aware that companies are advertising EDNP food products to them, participants often felt a sense of guilt for buying into such advertisements and not consuming the healthy foods that they “should” be. As health professionals, we should avoid tendencies to moralise the discussion around food and healthy eating, as these are often filled with shame, guilt, and fear tactics [89].

### 4.5. Limitations

Several limitations of this current study include the method of recruitment, where only individuals from within market research panels were invited to participate. The sample of participants were likely to be completing tertiary education and were skewed towards those more educated. Participants were generally studying and living with their family. Although this is typically representative of the Australian population [90], this may have meant these young adults were not in full control of purchasing and cooking main meals and their discussion may have been more around convenience or snack foods. We did not ask participants whether they were primarily in control of shopping and cooking meals and did not measure dietary intake, and thus we were unable to determine the level of control these young adults had over their meals. Differences in opinions amongst those from different cultures were not explored. There was a large attrition rate across the four weeks the online conversations ran, which may suggest this length of study may not be suitable for this age group and may have led to a bias in those completing being more interested in health. As participants were able to communicate with each other within each of the forums, the likelihood of participants displaying features of groupthink may be higher. Rather than sharing their own unique thoughts and experiences, participants may have sought unanimity, limiting the experiences they shared.

### 4.6. Implications for Future Practice

Future work should focus on identifying strategies to overcome the persuasive nature of EDNP food advertising both online and in traditional media. Going beyond the current food advertising regulations, which mainly focus on children only [41,42,49], would reduce the frequency and nature of EDNP foods being promoted. Concurrently, there should be an increase in the marketing of healthy food and evidence-based nutrition advice, particularly online and on social media. Healthy food advertising could increase its persuasive power by utilising some of the marketing appeals that caught young adults’ attention in these findings.

## 5. Conclusions

In conclusion, advertising of food products is widely prevalent in social media and traditional media for young adults. Commercial food advertising is noticed and often discounted because of its overt persuasion techniques. Young adults in this study were increasingly using protective measures such as ad-blockers to avoid covert advertising. However, the “weight” of covert advertising and persuasion is un-assessable because attention is not paid to natural, native, or embedded persuasion attempts. Young adults in this study were susceptible to advertising that appeals to pleasure-seeking (taste), cost, and convenience. Health professionals seeking to change food-related behaviours can consider widening the use of the marketing mix and advertising strategies to include appeals that are applicable to young adults’ lived experiences. For young adults to realise healthy eating behaviours, the online and in-person advertising environment must support the consumption of healthy foods.

## Figures and Tables

**Figure 1 nutrients-13-01934-f001:**
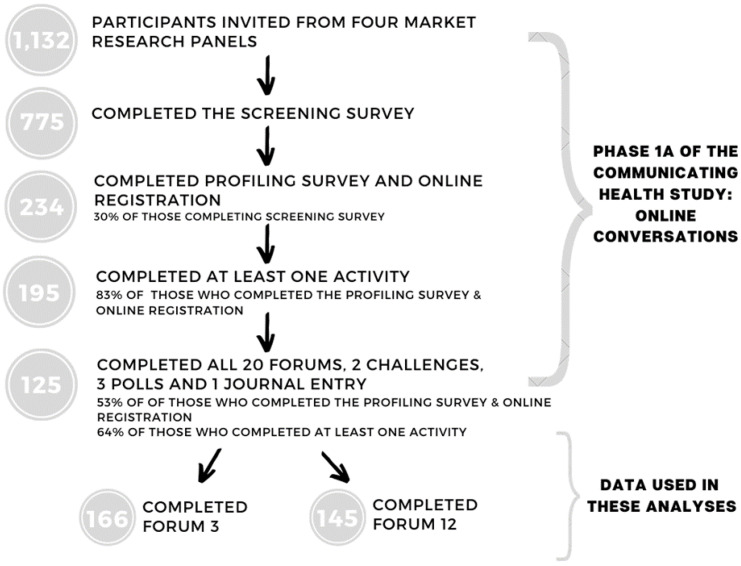
Communicating Health online conversations study flow diagram.

**Table 1 nutrients-13-01934-t001:** Discussion guide and logic of enquiry.

Forum	Discussion Guide	Logic of Inquiry
Forum 3: Ads about food	Over the course of this online community, let’s post here all the food related ads that we’ve noticed online over the recent weeks or that we’re noticing now and let’s discuss what caught our attention, what we like and don’t like... For any ad that is posted here by another member, please comment too and share whether you had noticed it before, what you like and don’t like.	Objective: Identify impactful food industry campaigns including triggers for engagement
Forum 12: The health ads we notice	Can you remember any health-related ads you’ve seen on social media? Let’s post all the health-related ads, articles, slogans, or anything that we noticed lately and discuss what comes to mind when we see these. Photos, links, screen grabs are all welcome.	Objective: Uncover triggers of interest

**Table 2 nutrients-13-01934-t002:** Participant characteristics of those completing either Forum 3 or Forum 12 or both (*n* = 166 participants).

Variable	Category	N Participants (% of Total) or Median (25th, 75th Percentile)
Gender identity	Female	101 (60.8%)
	Male	64 (38.6%)
	Non-binary/genderfluid/genderqueer	1 (0.6%)
Age (years)		21 (19, 23)
	18–21	92 (55.4%)
	22–24	74 (44.6%)
Location type *	Metro	133 (80.1%)
	Regional	33 (19.9%)
Language spoken at home	English	124 (74.7%)
	Language other than English	42 (25.3%)
Living arrangements †	Living with parents	80 (48.2%)
	My partner	35 (21.1%)
	Friend(s)/housemate(s)	28 (16.9%)
	Alone	19 (11.4%)
	Living with own child(ren)	17 (10.2%)
	Other family	17 (10.2%)
Dispensable weekly income ‡	Less than AUD 40	65 (39.2%)
	AUD 40–79	48 (28.9%)
	AUD 80–119	29 (17.5%)
	AUD 120–199	11 (6.6%)
	AUD 200–299	9 (5.4%)
	AUD 300 or over	3 (1.8%)
	I don’t wish to say	1 (0.6%)
Currently studying	Yes	111 (66.9%)
	No	55 (33.1%)
Level of current study (only completed by those who said they were currently studying)	High school, year 12	8 (4.8%)
	TAFE, college, or diploma	13 (7.8%)
	University (undergraduate course)	80 (48.2%)
	University (postgraduate course)	10 (6.0%)
Highest level of education completed (only completed by those who said they were not currently studying)	High school, year 10 or lower	1 (0.6%)
	High school, year 11	2 (1.2%)
	High school, year 12	12 (7.2%)
	TAFE, college, or diploma	23 (13.9%)
	University (undergraduate course)	15 (9.0%)
	University (postgraduate course)	2 (1.2%)
Main cultural identity	Caucasian (e.g., Australian, European)	130 (78.3%)
	East and South Asian (e.g., Chinese, Japanese, Vietnamese)	20 (12.0%)
	West Asian and Middle Eastern (e.g., Indian, Pakistani, Sri Lankan)	10 (6.0%)
	Aboriginal Australian	4 (2.4%)
	New Zealander	2 (1.2%)
Body mass index kg/m^2^ (calculated from self-reported height and weight)		23.8 (20.4, 27.5)
	Underweight (BMI < 18.5 kg/m^2^)	17 (10.2%)
	Healthy weight (BMI 18.5–24.9 kg/m^2^)	88 (53.0%)
	Overweight (BMI 25.0–29.9 kg/m^2^)	36 (21.7%)
	Obese (BMI ≥ 30.0 kg/m^2^)	25 (15.1%)

* Location question: “Please confirm where you live: 1. Sydney metro area; 2. Other New South Wales (regional/rural); 3. Melbourne metro area; 4. Other Victoria (regional/rural); 5. Brisbane metro area; 6. Other Queensland (regional/rural); 7. Adelaide metro area; 8. Other South Australia (regional/rural); 9. Perth metro area; 10. Other Western Australia (regional/rural); 11. Hobart metro area; 12. Other Tasmania (regional/rural); 13. Australian Capital Territory (Metro); 14. Northern Territory (regional/rural)”. † Participants could choose more than one response. ‡ Dispensable weekly income question: ‘During a normal week, how much money do you have to spend on yourself for recreational purposes?’

**Table 3 nutrients-13-01934-t003:** Summary of thematic analysis.

Theme	Summary
Theme 1—Promotion	These young adults “knew” they were being advertised to both online and in traditional media. Many participants utilised ad-blocking applications on social media to block out unwanted marketing. Advertisements that caught their attention were often visually appealing, bright, and included “yummy” energy-dense, nutrient-poor foods. Jingles, songs, and jokes stuck in the minds of these young adults. Seeing the same advertisement often was annoying but attention-grabbing and memorable.
Theme 2—Product	Participants mostly recalled advertisements for energy-dense, nutrient-poor and convenience foods. Advertisements for foods that the young adults already like and eat were attention-grabbing and memorable. Participants felt the advertising environment both online and in real-life hindered their ability to eat healthy and led to feelings of guilt when they were “tempted” by the unhealthy food advertised.
Theme 3—Price	Discounts and meal deals were one of the most attention-grabbing aspects of a food advertisement. Among participants, there were differing opinions over which was more expensive, energy-dense, nutrient-poor food convenience foods or cooking whole foods from scratch. When deciding which foods to purchase, convenience and price were key.
Theme 4—Place	Food advertisements were mainly seen on Facebook, YouTube, and Instagram. Traditional media (television, radio, and billboards) were a common place for young adults to see food advertisements. Participants preferred to buy foods in locations convenient to them, for example, at university or through food delivery.

## Data Availability

The datasets generated and/or analysed during the current study are not publicly available as consent was not provided by the participants to provide their responses outside of the study team but are available from the corresponding author on reasonable request.

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
