# Peer review of "Effects of Advertising: A Qualitative Analysis of Young Adults’ Engagement with Social Media About Food"

_nutrients, 2021, doi:10.3390/nu13061934_

Round 1

Reviewer 1 Report

Overall, this is an interesting paper on a topic of public health interest. However, the following points should be addressed.

The title is not reflective of the main findings of the study, and should be change. The first part should be removed

It seems like there was a large volume of data that was synthesized. However, the findings don’t really reflect this richness of data. Participant characteristics are presented in table 1; however most of the results are presented without any reference to this background info. It might not have been the aim of the paper to compare different groups; were there however some distinct differences between some groups based on the characteristics described? It is also not really possible to see which findings were overwhelmingly supported by participants; you mostly use “some participants….”, “often…”. Could you include some information in this regard?

What is also confusing is that, in your discussion, you bring the issue of the 4p framework. The categories in your results thus did not emerge from data, but from your previous knowledge of the framework, into which you fit your findings. How can you then argue that what you did is inductive thematic analysis? Please explain.

In your discussion, please reflect more on the implications of your findings for future practice

Author Response

Point 1: Overall, this is an interesting paper on a topic of public health interest. However, the following points should be addressed.

Response 1: Thank you for your review. We appreciate your feedback that has helped to improve the clarity of our paper.

Point 2: The title is not reflective of the main findings of the study, and should be change. The first part should be removed

Response 2: We have removed the first part and altered the title to “Effects of advertising: a qualitative analysis of young adults' engagement with social media about food”.

Point 3: It seems like there was a large volume of data that was synthesized. However, the findings don’t really reflect this richness of data. Participant characteristics are presented in table 1; however most of the results are presented without any reference to this background info. It might not have been the aim of the paper to compare different groups; were there however some distinct differences between some groups based on the characteristics described?

Response 3: The results were reported as the whole group rather than a comparison between these background or demographic characteristics as this was not an aim of this study. The participant characteristics were used to highlight the overall participant group who completed and were not used in analysis. We have added an explanation of this into the methods section.

“The analysis was conducted on the group of participants as a whole as making comparisons between different groups based on their characteristics was not the aim of this study.” (Lines 209-211)

Point 4: It is also not really possible to see which findings were overwhelmingly supported by participants; you mostly use “some participants….”, “often…”. Could you include some information in this regard?

Response 4: We did not quantify the amount of participants discussing each finding as we were not conducting content analysis and as we were asking open questions. We do not believe it is appropriate in our thematic analysis using an interpretivist paradigm to report percentages to avoid misleading conclusions or generalisations about the prevalence of opinions. We followed the methods for reporting from Patton 2014 (Patton MQ. 2014 Qualitative research & evaluation methods: Integrating theory and practice). The use of ‘some’ and ‘often’ are commonly used in qualitative research results to indicate that not the majority of participants mentioned this finding but it was mentioned across multiple participants, whereas ‘most’ indicates that a larger majority of participants mentioned this phenomena. All findings reported were mentioned by multiple participants.

Point 5: What is also confusing is that, in your discussion, you bring the issue of the 4p framework. The categories in your results thus did not emerge from data, but from your previous knowledge of the framework, into which you fit your findings. How can you then argue that what you did is inductive thematic analysis? Please explain.

Response 5: Thank you for pointing this out, we agree that it was currently not clear how this framework was incorporated into the research. We have now added into the results an explanation that this framework was not used in the analysis but was rather used to help explain the findings of the inductive analysis as it is a framework that the authors were previously aware of. 

“During triangulation of findings authors had similar themes related to the promotion of products and price. The authors believed these themes fit with the marketing mix or the 4P’s of marketing framework that is an established framework in marketing [61]. Therefore, the 4P’s framework was used to guide the discussion of the results from the participants. The four major themes represented therefore were: Theme 1- Promotion, Theme 2- Product, Theme 3- Price and Theme 4- Place.” (Lines 251-257)

Point 6: In your discussion, please reflect more on the implications of your findings for future practice

Response 6: We have now added a section to the discussion on implications for future practice.

“Future work should focus on identifying strategies to overcome the persuasive nature of EDNP food advertising both online and in traditional media. Going beyond the current food advertising regulations, which mainly focus on children only [41, 42, 49] would reduce the frequency and nature of EDNP foods being promoted. Concurrently, there should be an increase in the marketing of healthy food and evidence-based nutrition advice particularly online and on social media. Healthy food advertising could increase its persuasive power by utilising some of the marketing appeals that caught young adults’ attention in these findings.” (Lines 574-582)

Reviewer 2 Report

This work presents a remarkable literature review and a very adequate structure. The explanation of the data collection and analysis process is adequate. The results are clearly presented and very interesting. This aspect is in line with the discussions and conclusions.  The only thing that would be appreciated is the inclusion of some more recent citations (from the last 2 years) dedicated to advertising and food. Especially noteworthy are the works that have focused on healthy food and that have an interesting and abundant literature in recent years. Some of these works could be cited in the introduction.

Author Response

Point 1: This work presents a remarkable literature review and a very adequate structure. The explanation of the data collection and analysis process is adequate. The results are clearly presented and very interesting. This aspect is in line with the discussions and conclusions.  

Response 1: Thank you very much for reviewing our paper.

Point 2: The only thing that would be appreciated is the inclusion of some more recent citations (from the last 2 years) dedicated to advertising and food. Especially noteworthy are the works that have focused on healthy food and that have an interesting and abundant literature in recent years. Some of these works could be cited in the introduction.

Response 2: Thank you, we agree the introduction could be strengthened by including some more recent citations and so have added these to the introduction. 

“This is particularly apparent with children and therefore governments have at-tempted to mitigate the effect of unhealthy advertisements by introducing regulations [41, 42].” (Lines 89-91)

“Unhealthy food advertisements have been shown to have a significantly more positive response by adolescents compared to healthy food and non-food related advertisements  [49].” (Lines 111-112)

Reviewer 3 Report

Section 2.3

It seems that the authors only included 2 out of 20 forums? Can the authors elaborate on this.

Section 2.4

Was there any validation work on the thematic analysis?

Results

Interesting excerpts, though I'd like to invite the authors to create a summary highlight in the end of the subsection stating what they had found in that particular theme.

Discussion

Well written just needs to be subheaded accordingly based on the results that the authors had found as it is a bit cluttered at the moment. 

Author Response

Point 1: Section 2.3

It seems that the authors only included 2 out of 20 forums? Can the authors elaborate on this.

Response 1: Thank you for reviewing our paper and for your feedback. We only included these 2 forums as they were the only forums out of the 20 that were related to advertising which was the aim of the analysis for this paper. This is outlined on lines 199 to 202.

“These specific forums were chosen based on their discussion of food advertisements and were therefore the only ones that provided insight into the research question about attitudes and experiences relating to food advertising.”

Point 2: Section 2.4

Was there any validation work on the thematic analysis?

Response 2: We completed investigator triangulation as a form of validation of the thematic analysis where two authors code the data separately and then come together to compare codes and themes to validate the findings. We followed the methods for triangulation as a form of validation from Patton 2014 (Patton MQ. 2014 Qualitative research & evaluation methods: Integrating theory and practice). This is outlined on lines 222 to 226.

“Investigator triangulation was employed to enhance the rigor of the thematic analysis and as a form of validation [60] where two authors coded the forums separately and grouped their like codes before meeting to compare and contrast their coding and to come to a consensus before developing and refining themes.”

Point 3: Results

Interesting excerpts, though I'd like to invite the authors to create a summary highlight in the end of the subsection stating what they had found in that particular theme.

Response 3: Thank you for this suggestion. We have now added a summary table which outlines the main findings in each theme. See Table 3 Line 444.

Point 4: Discussion

Well written just needs to be subheaded accordingly based on the results that the authors had found as it is a bit cluttered at the moment.

Response 4: We have now added sub-headings and hope this improves the clarity of the discussion.

Round 2

Reviewer 1 Report

My concerns have been addressed.

Reviewer 3 Report

I'd like to thank the authors for addressing the comments.